# Melittin Tryptophan Substitution with a Fluorescent Amino Acid Reveals the Structural Basis of Selective Antitumor Effect and Subcellular Localization in Tumor Cells

**DOI:** 10.3390/toxins14070428

**Published:** 2022-06-22

**Authors:** Yonghui Lv, Xu Chen, Zhidong Chen, Zhanjun Shang, Yongxiao Li, Wanting Xu, Yuan Mo, Xinpei Wang, Daiyun Xu, Shengbin Li, Zhe Wang, Meiying Wu, Junqing Wang

**Affiliations:** 1School of Pharmaceutical Sciences, Shenzhen Campus of Sun Yat-Sen University, Shenzhen 518107, China; lvyh9@mail2.sysu.edu.cn (Y.L.); chenx589@mail2.sysu.edu.cn (X.C.); chenzhd9@mail2.sysu.edu.cn (Z.C.); shangzhj@mail2.sysu.edu.cn (Z.S.); liyx356@mail2.sysu.edu.cn (Y.L.); xuwt27@mail2.sysu.edu.cn (W.X.); moyuan@mail2.sysu.edu.cn (Y.M.); wxpsybella@163.com (X.W.); xudy8@mail2.sysu.edu.cn (D.X.); lishb28@mail2.sysu.edu.cn (S.L.); 2Department of Pathology, The Eighth Affiliated Hospital, Sun Yat-sen University, Shenzhen 518033, China

**Keywords:** melittin, Trp, Dap, AMCA, fluorescence, alpha helix, nucleolus

## Abstract

Melittin is a membrane-active peptide with strong anticancer activity against various cancers. Despite decades of research, the role of the singular Trp in the anticancer activity and selectivity of melittin remains poorly understood. Here, we propose a theranostic solution based on the substitution of Trp19 with a noncanonical fluorescent amino acid (Dap^AMCA^). The introduction of Dap^AMCA^ residue in melittin stabilized the helical structure of the peptide, as evaluated by circular dichroism spectra and molecular dynamics simulations. In vitro hemolytic and anticancer activity assays revealed that introducing Dap^AMCA^ residue in melittin changed its mode of action with the cell membrane, resulting in reduced hemolytic toxicity and an improved the selectivity index (SI), with up to a five-fold increase compared to melittin. In vitro fluorescence imaging of Dap^AMCA^-labeled melittin (MEL^FL^) in cancer cells demonstrated high membrane-penetrating activity, with strong nuclear and nucleolar localization ability. These findings provide implications for novel anticancer therapies based on Trp-substituted designs and nuclear/nucleolar targeted therapy.

## 1. Introduction

Melittin is the primary toxic component of bee venom, and is produced by the European honeybee, *Apis mellifera* [1]. It is a 26-mer amphiphilic peptide (NH_2_-GIGAVLKVLTTGLPALISWIKRKRQQ-CONH_2_) with a net charge of +5 at pH 7.2, consisting of two α-helical segments punctuated by a proline residue; the hydrophobic residues are mainly distributed in the N-terminal region (1–20), whereas the hydrophilic residues are localized in C-terminal tails (21–26). Melittin has been shown to have strong antimicrobial, antiviral, and anticancer activity both in vitro and in vivo [2,3,4,5,6]. However, the clinical therapeutic applicability of melittin has been impeded by its nonspecific cytotoxicity and hemolytic activity. The non-selective cytolytic effect of melittin is mainly associated with active pore formation through its disruption of the cell membrane [5,7].

Tryptophan (Trp, W) is one of the important residues of membrane proteins, which primarily act at the lipid–water interface and facilitate their membrane anchoring, organization, and function in a lipid bilayer [8]. Previous findings have shown that a unique tryptophan residue in melittin is crucial for its hemolytic activity [9,10]. Selective labeling or substituting of this Trp residue in melittin is a key method for studying the interaction of melittin with membranes. The insertion and orientation of melittin in the lipid membrane have previously been monitored by substituting L9 and W19 residues in the melittin sequence with a dual-fluorescence noncanonical amino acid (AFaa) [11]. Later, an alternative method replaced the W19 residue with b-(1-azulenyl)-L-alanine (AzAla) in order to provided polarity-independent fluorescence properties without introducing significant structural or functional impacts [12]. Furthermore, a recent study demonstrated that the selective ruthenium labeling of the Trp residue in melittin could alter peptide–membrane interaction patterns, leading to a reduction in hemolytic activity [13]. Although these site-specific approaches have been successfully applied to investigate melittin–membrane interactions, the role of Trp in the anticancer activity and selectivity of melittin remains poorly understood.

In the current study, we report the in situ visualization and investigation of fluorescently-labeled melittin in cancer cells based on the replacement of the Trp residue with Dap^AMCA^, a (2S)-2,3-diaminopropanoic acid (Dap) side chain-labeled 7-Amino-4-methyl-3-coumarinylacetic acid (AMCA) (Figure 1). AMCA was selected due to its small fluorophore moiety and high fluorescence yield, with λex/λem at 345–355 nm/440–460 nm. This allowed us to visualize the subcellular localization and distribution of melittin with minimal conformational disturbance upon Trp−19−Dap^AMCA^ mutation. Here, we aimed to use the fluorescent noncanonical amino acid Dap^AMCA^ as an in situ probe marker for monitoring melittin action at the molecular and cellular levels. We initially analyzed and compared the conformational stability of the secondary structure in native and mutant fluorescent melittin (MEL^FL^). The hemolytic activity of MEL^FL^ was then determined and compared with native melittin. Following structural and in vitro safety analysis, the human and mouse cancerous cell lines MCF-7 and Hepa 1-6 were investigated as in vitro model systems to evaluate the therapeutic effect on the cancer cell lines in order to probe melittin’s cellular distribution and active site.

## 2. Results and Discussion

### 2.1. The Structural Effect of Dap^AMCA^ Substitution on Melittin

Melittin and its Dap^AMCA^ substitution analog were synthesized via solid-phase peptide synthesis (SPPS) and characterized by HPLC and mass spectrometry (Appendix A, Appendix A). In order to investigate the intrinsic helical propensities of melittin and its analog, we measured the circular dichroism (CD) spectra of each peptide at 0.125, 0.25, and 0.5 mg·mL^−1^ in an aqusolvent with the presence of 2,2,2-Trifluoroethanol (TFE). TFE is a commonly used cosolvent to mimic the membrane environment and stabilize the α-helicity of polypeptides [14]. The obtained CD spectra of each peptide consist of pronounced double negative peaks at 222 and 208 nm and a positive band with a peak at~190–193 nm (Figure 2A,B). The minimum ellipticity values observed at the characteristic peaks for melittin and MEL^FL^ indicate the formation of an α-helical structure. In order to estimate the α-helical content of proteins, two equation models that adopt residue ellipticity at 222 and 208 nm were applied (Table 1). This shows that an increase in the concentrations of each peptide (0.125–0.5 mg·mL^−1^) resulted in a moderate reduction in α-helical content. Notably, the calculation results based on the ellipticity of [θ]_208_ are higher than those found on [θ]_222_. For example, 0.5 mg·mL^−1^ of melittin exhibited a mean α-helicity of~76%, while MEL^FL^ displayed α-helix contents of~39%, respectively. The obtained α-helicity of melittin here is in good agreement with previous results obtained in similar studies [15,16], and the trend and internal relationship between the two settlement results remain highly consistent. Because the CD spectroscopy method is based on the signature of two transition mixtures of the peptide bond in the backbone (amide chromophore N–C = O occurs at 190 and 220 nm), contributions from side-chain isopeptide bond and chromophore (AMCA) may alter the overall CD signal if they are constrained in asymmetric geometries [17,18]. In this respect, the lower ellipticity values found at 222 and 208 nm for MEL^FL^ relative to melittin reflect a ~50% loss of ellipticity signal, suggesting that Dap^AMCA^ substitution at W19 residue induces either a regional conformation disorder, or signal interference from the isopeptide bond and chromophore on the side-chain of Dap^AMCA^ in the melittin.

In order to capture the peptide folding behaviors and obtain further insight into changes in the secondary structure of two peptides, we carried out all-atom molecular dynamics (MD) simulations of melittin and MEL^FL^ using the GROMACS software package (https://www.gromacs.org, accessed on 9 April 2021) [19,20]. The input structures were adapted from the crystal structure of the melittin monomer from the PDB bank (PDB code: 2MLT). The initial conditions of the MD simulations, including temperature and solvent, were selected to simulate the experimental conditions. Finally, a total of 2 μs MD simulations of native melittin and MEL^FL^ were performed at 310 K in 50% (*v/v*) TFE/water cosolvent systems. Figure 2C shows the root mean square deviation (RMSD) with respect to the initial structure of each polypeptide as a function of time. During the 1 μs trajectory, both polypeptides remain stationary with respect to their initial conformation, and each system reaches a stable folded state with RMSD at~2.4-2.6 Å after~500 ns. Notably, MEL^FL^ experienced fewer fluctuations in RMSD value than melittin. The lower average RMSD score obtained from MEL^FL^ indicates its smaller conformational variations and higher structural rigidity compared to melittin (Figure 2C). A more detailed analysis of the root mean square fluctuation (RMSF) profile was determined considering per-residue flexibility. We found that the terminal regions underwent large relative fluctuations in the conformations of both systems throughout MD simulation (Figure 2D).

In contrast, the helical architecture of the N-terminal hydrophobic segment (G1-I17 residues) in both peptides is comparatively stable, while the C-terminal hydrophilic tail (S18-Q26) is relatively unstable due to more side-chain rotatable bonds in Arg, Lys, and Gln residues, which yielded higher structural flexibility in both polypeptides [21]. This shows that the Trp19 in melittin is substantially affected by this flexible domain [22]. However, Trp19 exhibited ~2X higher RMSF values in trajectory than the Dap^AMCA^19 in MEL^FL^, which displays fewer conformational fluctuations in C-terminal residues (Figure 2D,E). We speculate that the introduction of Dap^AMCA^19 at Trp19 in melittin can effectively constrain the C-terminal domain in an α-helical conformation (Figure 2F). In order to further compare the stability of the α-helical structure between melittin and MEL^FL^, the α-helical content of each peptide was calculated as a percentage from the trajectories (Figure 2G). Both peptides’ conformation reached a steady state in the last 500 ns, and MEL^FL^ exhibited a higher α-helical content than melittin, with an average value of~90% versus 85%, respectively, indicating that the Trp−19−Dap^AMCA^ mutation increased the α-helix propensity of the MEL^FL^ peptide in the 2,2,2-trifluoroethanol/water cosolvent system. This result further confirms that Dap^AMCA^ may provide robust geometrical hindrance that restricts the conformational variations of adjacent residues.

Changes in the accessibility of peptides to solvent were determined by calculating the solvent-accessible surface area (SASA). From Figure 2H, MEL^FL^ showed comparatively higher SASA values than the native type after the systems reached a steady state. The increased SASA could be attributed to the larger side-chain moiety of Dap^AMCA^ compared to Trp. These observations found that the conformational stability of the melittin could be improved by the Trp−19−Dap^AMCA^ mutation, despite the fact that the introduction of Dap^AMCA^ with an isopeptide bond may have caused the α-helix content obtained from the CD spectra of MELs to be quite different from the theoretically predicted values.

Previous studies on the self-aggregation dynamics of melittin have reported that it forms a tetrameric helical structure in aqueous solution [22,23]. The melittin tetramer binds directly to the phospholipid membrane, resulting in the formation of stable toroidal pores and inducing strong membrane lytic activity [24]. Thus, we investigated the differences in the stability of the tetrameric state in melittin and MEL^FL^ in aqueous solution through 500 ns MD simulations. The structure change of the tetramer system with respect to the initial state was monitored by plotting RMSD trajectories (Figure 2I,J). For melittin, we observed that the RMSD value of the melittin tetramer varied between 2 and 4 Å over 500 ns; the RMSD of the four melittin chains fluctuated with varying degrees between 4 and 5 Å, and the overall trajectory of the tetramer tended to be stable (Figure 2I). In contrast, the simulated RMSD for MEL^FL^ tetramer increased to above 6 Å after about 100 ns. This appears to be a loss of overall stability of the MEL^FL^ tetramer due to a relatively large increase in the RMSD of one of the single strands (chain 2), particularly in the last 100 ns of the trajectory (Figure 2J). The comparison between snapshots of the initial and final states of the tetramers shows that the MEL^FL^ tetramer experienced greater structural change than the melittin tetramers. Because melittin adopts a tetrameric helical structure with high ionic strength under natural aqueous conditions [22,25], these results suggest that the Trp−19−Dap^AMCA^ mutation could result in loss of the tetrameric state of MEL^FL^, and may subsequently alter its membrane accessibility.

### 2.2. The Effects of Dap^AMCA^ Introduction on Melittin Hemolytic/Cytolytic and Cytotoxic Activity

A rabbit-erythrocyte (RBC) hemolytic assay was carried out to determine the membrane disruption activities of melittin and MEL^FL^. For both melittin and MEL^FL^, cells lysed with 2% Triton X-100 were used as the positive control, while cells treated with PBS buffer solution were assayed as a negative control. As measured by increased absorbance of RBC released hemoglobin, we found that melittin achieved total (100%) RBC hemolysis at a concentration of 8 μg·mL^−1^. In comparison, the cells treated with an equivalent concentration of MEL^FL^ reached less than 7% hemolysis. The hemolytic activities of melittin and its analog were determined at a hemolytic concentration of 50% (HC_50_), which is the most commonly used indicator of drug toxicity [26]. As shown in Figure 3A, The HC_50_ values of hemolysis for melittin and MEL^FL^ were 3 and 40.3 μg·mL^−1^, respectively. Compared to native melittin, Trp−19−Dap^AMCA^ substitution led to a substantial (13-fold) reduction in hemolytic activity. This observation is in good agreement with the MD simulation as well as with previous experimental results [13,27]. It strongly suggests that the Trp residue of melittin is crucial for inducing hemotoxicity. Apparently, the introduction of the Dap^AMCA^19 residue changes the mode of interaction of melittin with RBC membranes, resulting in reduced pore-forming activity [27,28,29].

In order to investigate differences in the morphological changes of cancerous and normal cell lines upon treatment with melittin and MEL^FL^, real-time observations with images captured using optical microscopy were performed at multiple time points. Melittin (10 μg·mL^−1^) and MEL^FL^ (30 μg·mL^−1^) were added to three cell lines (MCF-7, Hepa 1-6, HUVEC) and cell morphological characteristics were observed over time. Before treatment, each cell type maintained a regular spindle or flat shape and a homogeneous membrane attached to the bottom of the flask, showing an adherent morphology (Figure 4, at 0 min). The melittin-treated cells exhibited noticeable morphological changes within 1 min, and all three cell lines transformed from a normal adherent morphology to a swollen spherical shape within 15 min, with cell membranes accompanied by blisters that continued to swell (Figure 4A). Similar morphological transitions and progressive loss of basal adhesion were observed in MEL^FL^-treated cancer cells (Figure 4B). However, in contrast to the melittin-treated normal human endothelial cells (HUVEC), the relevant changes in membrane morphology were not observed in MEL^FL^-treated HUVEC. In addition, HUVEC cells maintained consistent morphology relative to the initial state during treatment. These observations show that the Trp19 mutation of melittin leads to differences in cytolytic activity between normal and cancer cells, suggesting higher selectivity of MEL^FL^ in cancer treatments.

Based on the hypothesis that the anticancer potential of melittin might be enhanced by reduced nonspecific membrane interactions, we evaluated the therapeutic activity of each polypeptide in human breast cancer MCF-7 cells and murine hepatoma Hepa 1-6 cells for 24 h incubation with the half-maximal inhibitory concentration (IC_50_) values reported in Table 2. Both peptides showed a concentration-dependent cytotoxic effect against MCF-7 and Hepa 1-6 cells (Figure 3B,C). As expected, melittin efficiently inhibited cell proliferation, with an IC_50_ of 10.22 and 6.39 μg·mL^−1^ in MCF-7 and Hepa 1-6 cells, respectively. In contrast, the MEL^FL^ exhibited moderate cytotoxicity, with about three-fold higher IC_50_ values (IC_50_ = 28.01 μg·mL^−1^ for MCF-7 and 22.92 μg·mL^−1^ for Hepa1-6) relative to native melittin (Table 2). This confirms that the nonspecific cytotoxic effect of melittin is mainly due to its membrane pore disruption mechanisms [30,31,32]. This is reflected in the steeper inhibition curve of melittin compared to that of MEL^FL^ (a certain concentration threshold leads to pore formation) [32]. Although MEL^FL^ displays attenuated cytotoxicity against cancerous cells, it takes advantage of selective anticancer activity, as indicated by the selectivity index (SI, HC_50_/IC_50_ ratio) for cancer cells versus RBC (Table 2) [33]. MEL^FL^ showed an SI > 1, four to five times more favorable than melittin (SI = 0.29 for MCF-7 and 0.47 for Hepa 1-6). This increase in the SI window for MEL^FL^ reveals more desirable overall antitumor effects with a wider margin of safety as compared to melittin [34].

In order to determine whether apoptosis contributed to the observed anticancer proliferation, we performed annexin V-fluorescein isothiocyanate/propidium iodide (annexin V-FITC/PI) apoptosis assays for melittin/MEL^FL^-treated MCF-7, Hepa 1-6, and HUVEC cells, followed by flow cytometry analysis. Cells were treated with IC_50_ of melittin and 30 μg·mL^−1^ (approximately 3X IC_50_ of melittin) of MEL^FL^ for 2 h each and stained with annexin V-FITC and PI. A typical flow cytometric dot plot is shown in Figure 5, divided into four quadrants as necrotic/late apoptotic cells (Q2, FITC+/PI+), early apoptotic cells (Q3, FITC+/PI-), and viable cells (Q4, FITC-/PI-). In the case of MCF-7 cells, we observed melittin-induced total apoptosis in 28.13% (early apoptosis = 5.16% and late apoptosis = 22.97%) of cells. The cells treated by MEL^FL^ showed a slightly lower total apoptosis rate (24.18%), 4.55% for early apoptosis and 19.63% for late apoptosis. In addition, the incubation of Hepa 1-6 cells with melittin and MEL^FL^ resulted in 36.6% and 34.34% of total apoptosis (Q2 + Q3), respectively. For HUVEC cells, it is noteworthy that the melittin-treated cells exhibited apparent apoptotic effects, with 28.87% total apoptosis, while only 9.17% of total apoptosis occurred in MEL^FL^-treated cells. This result suggests that MEL^FL^ is more prone to induce apoptosis in cancer cells than in normal cells.

In order to further determine the differences in apoptotic effects of melittin and MEL^FL^-induced in different cell types, a statistical analysis was performed using one-way ANOVA and Student’s *t*-test. Among the three cell lines in which melittin induced apoptosis (Figure 5B), Hepa 1-6 cells were more sensitive to melittin exposure compared to MCF-7 and HUVEC cells, while no difference was observed in the apoptotic effect of melittin on MCF-7 and HUVEC cells. Comparatively, MEL^FL^ clearly induced apoptosis in cancer cells; however, the effect of MEL^FL^ in inducing apoptosis of HUVEC cells was significantly lower. In addition, in the same cell line comparison (Figure 5C) it was found that melittin induced more pronounced apoptosis than MEL^FL^ in MCF-7 cells, while for Hepa 1-6 cells, melittin and MEL^FL^ showed comparable apoptosis-inducing effects. However, HUVEC cells treated with melittin showed significant apoptosis (roughly three-fold higher) compared with those treated by MEL^FL^. Together, these findings indicate that although melittin and MEL^FL^ demonstrate comparative apoptotic effects on cancer cells, the apoptotic effect of melittin on the normal cell line is more pronounced than in those treated by MEL^FL^, which is in line with the observations from the study of cell membrane morphological changes. Collectively, our data indicate that the replacement of Trp with Dap^AMCA^ can greatly reduce the RBC membrane-binding activity of melittin and thereby achieve an increase in SI that is expected to be therapeutically beneficial in anticancer treatment.

### 2.3. In Situ Visualization and Morphological Evaluation of MEL^FL^ in Cancerous Cell

In order to study the membrane activity and localization of MEL^FL^ in MCF-7 and Hepa 1-6 cells, cultured cells were stained with Nuclear Green™ LCS1 (green) and incubated with MEL^FL^ (blue) at 30 μg·mL^−1^ for 2 h and examined via confocal laser scanning microscopy (CLSM) characterization. The obtained CLSM images show that cell nuclei were clearly identified by LCS1 staining and MEL^FL^s was widely distributed on the plasma membrane, indicating that MEL^FL^ integrated into cell membranes (Figure 6). In addition, the overlay fluorescence images (526 + 450 nm) demonstrate the intracellular penetration of MEL^FL^, and the blue fluorescence signal predominantly localizes at the cytoplasm and nucleoli, with faint nuclei dispersion. The zoomed-in view of the ROIs for MCF-7 cells shows that LCS1-stained nuclear DNA heterochromatin is associated with the nuclear lamina, and bright blue spots (MEL^FL^) within the nuclei overlap with nucleoli, a higher mass density region with darkened LCS1 signals. The colocalization of MEL^FL^ with nucleoli can be readily confirmed under bright-field microscopic observation, in which the nucleolus appears as a dense body surrounded by dense fibrillar components [35]. Similar observations were found in Hepa 1-6 cells. These results reveal that nucleoli fluorescence originates from the binding of MEL^FL^ with ribosomal RNA (rRNA) and proteins of nucleoli in the nucleus. The specific accumulation of AMCA signals in the nucleolus of MEL^FL^ could be attributed to the C-terminal basic cluster KRKR, which contains a sequence similar to nuclear localization signals (NLS), a short signal peptide responsible for the nuclear transport of large biomolecules [36,37,38]. These findings suggest that melittin may exert an anticancer effect through a new pathway associated with interference in ribosomal biogenesis [39].

## 3. Conclusions

In summary, we demonstrated a facile approach for substitution of Trp19 residues in melittin with a fluorescent-labeled amino acid (Dap^AMCA^), which effectively reduced nonspecific cytolysis while retaining cytotoxic activity in cancer cell lines. The introduction of Dap^AMCA^ residue in melittin reinforced the peptide’s conformational stability via sidechain-assisted rigidity enhancement of the C-terminal backbone. However, this resulted in the loss of melittin’s tetrameric structure and mode of action with respect to the cell membrane, which significantly reduced hemolytic activity and markedly improved the selectivity index (SI) compared to melittin. Furthermore, in vitro high-resolution fluorescent imaging of Dap^AMCA^-labeled melittin (MEL^FL^) demonstrated membrane-binding ability and high membrane-penetrating activity, with strong nuclear and nucleolar localization ability. Our findings clearly show that Trp19 plays a key role in mediating nonspecific cytolytic activity. We suggest that Trp19 displacement in melittin may be considered a potential strategy for improving anticancer selectivity and nucleolar targeted therapy. We hope that the described method find application in other biochemical and biomedical studies regarding membrane-active peptides. Nevertheless, the intracellular mode of action of melittin remains to be further explored in seeking to address the limitations of melittin in cancer treatment.

## 4. Materials and Methods

### 4.1. Materials and Regents

Chemical reagents such as trifluoroethanol (TFE, >99%), dimethyl sulfoxide (DMSO, >99%), and all side-chain protected amino acids (99%) were purchased from Macklin (Shanghai Macklin Biochemical Co., Ltd., Shanghai, China). Ultra-pure water was obtained by a Millipore Milli-Q Integral 3 Ultrapure water system (Billerica, MA, USA). Thiazolyl blue tetrazolium bromide (MTT), Triton X-100, and Annexin V-FITC/PI Apoptosis Detection Kit were purchased from the Beyotime Biotech Company (Shanghai, China). New Zealand rabbit blood was purchased from Guangzhou Hongquan Biotechnology Co., Ltd., Guangzhou, China. The cell culture-related reagents were purchased from Gibco (New York, NY, USA). All reagents were used as received from commercial suppliers without further purification.

### 4.2. Polypeptide Synthesis

Melittin (MEL) and its analog (MEL^FL^) were synthesized using the solid-phase synthesis method with modifications that follow the process previously reported in the literature [40]. As described in Figure 1, the crude peptide was purified using a Waters Alliance 2695 HPLC System with 2487 Dual Absorbance Detector (Waters Alliance, Milford, MA, USA) on a ZORBAX SB-C18 (Agilent, Santa Clara, CA, USA) column in order to obtain a purity over 95%. A gradient of 5–70% mobile phase consisting of solvent A (0.1% TFA in 100% acetonitrile) and solvent B (0.1% TFA in 100% water) was used as the eluent over 20 min. Ten microliters of the sample were analyzed at 220 nm at a flow rate of 1 mL/min. The retention time of the target peak was about 11 min for MEL and 17.5 min for MEL^FL^. The determination of molecular weight was carried out using a Waters ZQ electrospray ionization mass spectrometer (Waters Alliance, Milford, MA, USA). The collection of the target peak was dried with nitrogen flow and then lyophilized under a vacuum desiccator. The lyophilized powder of peptides was finally stored at −20 °C before use.

### 4.3. Cell Lines and Cell Culture

Mouse liver cancer cells (Hepa 1-6), human breast cancer cells (MCF-7), and human umbilical vein endothelial cells (HUVEC) were purchased from Cell Bank of Shanghai, Chinese Academy of Science (CAS, Shanghai, China). The cells were cultured in high-glycemic Dulbecco’s Modified Eagle Medium (DMEM) supplemented with 10% fetal bovine serum (FBS) and 1% penicillin and streptomycin. Cells were cultured in an incubator (Esco Micro Pte. Ltd., Changi, Singapore) in an atmosphere of 5% CO_2_ and 90% relative humidity at 37°. In order to maintain good growth conditions, culture operations were carried out in accordance with the ATCC’s instructions. The daily growth status of the cells was observed under a binocular phase contrast microscope, and the cells growing in the logarithmic phase were used in related experiments.

### 4.4. Secondary Structure Research

The secondary structure of polypeptides was investigated with a Jasco 720 spectrometer between 190–250 nm at 37 °C. To obtain peptide solutions at serial concentrations (0.5, 0.25, 0.125 mg·mL^−1^), the stock solution of peptides was obtained by dissolving an appropriate amount of peptide powder in 50% trifluoroethanol (TFE) aqueous solution (*v*/*v*, TFE/Water), then performing a half-fold dilution. The prepared sample was transferred into a quartz cuvette (1 mm light path). The spectrum in the wavelength range of 190 nm to 250 nm was recorded (step 1 nm) at 37 °C. Fifty percent of TFE buffer was measured as a control. Molar residue ellipticities (deg cm^2^ dmol^−1^) were calculated from raw ellipticities (mdeg). The percentage of peptide α-helix content was obtained according to two empirical equations: (1) α helix% = (−[Ѳ]_208_ + 4000)/(33,000–4000) × 100, where [Ѳ]_208_ is the average residue molar ellipticity at 208 nm [41]; and (2) α-helix% = ([Ѳ]_222_ + 3000)/36,000 × 100, where [Ѳ]_222_ is the average residue molar ellipticity at 208 nm [42].

### 4.5. Hemolysis Assay

The hemolytic activity of the peptides was investigated using the method described by Jing Dong et al. [43]. One milliliter of sterile PBS was added to an equal volume of New Zealand rabbit blood, after which the red blood cells were pelleted by centrifugation (7000 rpm, 3 min, Dalong D1008, Beijing Dalong Xingchuang Experimental Instrument Co., Ltd., Beijing, China) and the supernatant was discarded. The obtained red blood cell pellet was resuspended in 2 mL sterile phosphate buffer saline (PBS) and then centrifuged again under the same conditions to discard the supernatant. The above process was repeated four or five times until uniform red blood cells were obtained, then 300 μL of 4% erythrocyte suspension and an equal volume of peptide gradient solution were uniformly mixed to obtain final peptide concentrations of 0, 1, 2, 4, 8, 16, 32, 64, and 128 μg·mL^−1^. Triton X-100 at a concentration of 2% was used as a positive control. The suspension of peptide (or Triton X-100) and red blood cells was incubated for 3 h at 37 °C. Then, the sample was centrifuged (7000 rpm, 3 min) and 100 μL/well of supernatant was added to a 96-well plate. The hemolysis rate of the red blood cells was analyzed by measuring the absorption at 540 nm through a SpectraMax i3 microplate reader (Molecular Devices, Sunnyvale, CA, USA). The hemolysis rate was obtained based on the following formula: hemolysis ratio% = (OD_t_ − OD_b_)/(OD_c_ − OD_b_) × 100; OD_t_, OD_c_, and OD_b_ denote the optical density value of the experimental group, the positive control group, and the blank group, respectively.

### 4.6. Cytotoxicity Investigation

The cytotoxicity of MEL and MEL^FL^ was determined by MTT assays. The cells grown in the logarithmic phase were seeded into 96-well plates at 1 × 10^4^ per well and then incubated for 24 h to adhere to the wall. After that, the medium was replaced with a solution containing different concentrations of MEL or MEL^FL^, which were prepared with fresh media. After incubating for an additional 4 h, the medium was removed. Then, 150 μL of dimethyl sulfoxide (DMSO) was added to each well and shaken at room temperature for 10 min to fully dissolve the formazan crystals. The signal was recorded at 490 nm with the SpectraMax i3 microplate reader (Molecular Devices, Sunnyvale, CA, USA). The cell viability percentage was obtained according to the following formula: cell viability % = (OD_t_ − OD_b_)/(OD_c_ − OD_b_) × 100; OD_t_, OD_c_, and OD_b_ denote the optical density value of the experimental group, the positive control group, and the blank group, respectively.

### 4.7. Optical Microscopy

The visualization of cell membrane lysis was carried out as described in previous studies [31]. Briefly, cells were seeded at 2 × 10^4^ cells per well in a 24-well plate overnight in complete DMEM medium. Cells were studied using optical microscopy (Nikon Eclipse Ti2 Inverted Microscope, Nikon instruments Co., Ltd., Shanghai, China) to visualize membrane changes resulting from the addition of 10 μg·mL^−1^ MEL or 30 μg·mL^−1^ MEL^FL^ in PBS over a 15-min time course.

### 4.8. Flow Cytometry

Apoptosis analysis was performed using an Annexin V-FITC/PI Apoptosis Detection Kit (Beyotime Inc., Shanghai, China) according to the manufacturer’s instructions. MCF-7, Hepa 1-6, and HUVEC were seeded into a 6-well plate and cultured in Dulbecco’s Modified Eagle Medium (DMEM) complete medium at 1 × 10^5^ per well at 37 °C and 5% CO_2_ overnight to fully adhere to the wall. Subsequently, medium containing melittin at 10 μg·mL^−1^ and MEL^FL^ at 30 μg·mL^−1^ was added. The cells were incubated for an additional 120 min. The negative control was obtained by culturing with peptide-free medium. The adherent cells were washed with sterile PBS and digested with 0.25% EDTA-free trypsin, then combined with the medium containing suspended cells, centrifuged for 3 min (7000 rpm, Dalong D1008, Beijing Dalong Xingchuang Experimental Instrument Co., Ltd.), and the supernatant was discarded. The cell pellet was resuspended in 500μL 1X binding buffer, then Annexin V-FITC and PI were added, respectively, and the solution was incubated in the dark for an appropriate length of time. Samples were analyzed by BD CytoFLEX. Student’s *t*-test and one-way ANOVA tests were used to assess statistical significance in apoptosis between MEL and MEL^FL^ for MCF-7, Hepa 1-6, and HUVEC. The data were expressed as the mean ± standard deviation with *n* = 3.

### 4.9. Confocal Fluorescence Microscopy

The in vitro fluorescence characteristics of MEL^FL^ were studied by confocal laser scanning imaging (CLSM, Carl Zeiss LSM 880; Carl Zeiss AG, Oberkochen, Germany). MCF-7 and Hepa 1-6 cells were seeded in a glass-bottomed cell culture dish at 5 × 10^4^ per well and incubated at 37 °C and 5% CO_2_ for 24 h to fully adhere to the wall. Fresh medium solution containing 30 μg·mL^−1^ peptides was added to the dish and the incubation was continued for 2 h. The cells were washed three times with pre-chilled sterile PBS, then nuclear green LCS1 was used for nuclear staining. An appropriate amount of anti-fluorescence quenching mounting solution was added prior to imaging. The emission wavelengths of nuclear green LCS1 and AMCA-labeled melittin derivatives are 526 nm and 450 nm, respectively.

### 4.10. Molecular Dynamics Simulations

The crystal structure of melittin was obtained from Protein Data Bank (PDB ID: 2MLT). The melittin topology file was generated using the GROMACS tool [44]. The CHARMM36 force field was applied in this work [45]. The Dap^AMCA^ residue of the mutants was parameterized using SwissParam (https://www.swissparam.ch/, accessed on 9 April 2021), and the topology parameters were incorporated into the melittin topology file [46]. Monomers and tetramers of melittin and MEL^FL^, respectively, were simulated for a total of four systems. The protein was placed in the cubic box in all simulations. The minimum distance from the protein to the box was set as 1.2 nm. Different solvation schemes were applied to melittin monomers and melittin tetramers in order to study their properties in different environments. For the simulation of melittin monomer, the system was solvated using transferable intermolecular potential 3 points (TIP3P) water and trifluoroethanol. Six chloride ions were added to the box to neutralize the system by replacing water molecules. For the simulation of melittin tetramer, the system was solvated using TIP3P water molecule model, and 0.145 M NaCl was added to the system. Energy minimization was performed to optimize the system using the steepest descent algorithm. A 1-ns run was performed in the NPT ensemble, and harmonic position restraints were applied to all heavy atoms. A 1000 ns production simulation of melittin monomer and a 500 ns production simulation of melittin tetramer were then run. The temperature of the system was set at 310 K and the pressure was set at 1 bar using the v-rescale algorithm and the Parrinello–Rahman algorithm. All hydrogen bonds were constrained using the LINCS algorithm [47]. Both the Van der Waals cut-off and Coulomb cut-off were set to 1 nm. Particle-mesh Ewald was applied to calculate long-range Coulomb interactions [48]. All trajectory analyses were performed using GROMACS (https://www.gromacs.org, accessed on 9 April 2021). Root mean square deviation (RMSD), root mean square fluctuation (RMSF), solvent accessible surface area (SASA), and α-helix percentages were calculated. The α-helix percentage represents the proportion of residues that form the alpha helix among all residues. The central structure of the trajectory was calculated by diagonalizing the RMSD matrix. The central structure had the smallest average RMSD in the cluster compared to other structures.

### 4.11. Statistical Analysis

The experiment was repeated three times in parallel. The results are expressed as mean ± SD. All experimental data were analyzed using Microsoft Excel software and GraphPad Prism8.4.3 (GraphPad Inc., San Diego, CA, USA). A *p*-value less than 0.05 was considered to be statistically significant.

## Figures and Tables

**Figure 1 toxins-14-00428-f001:**
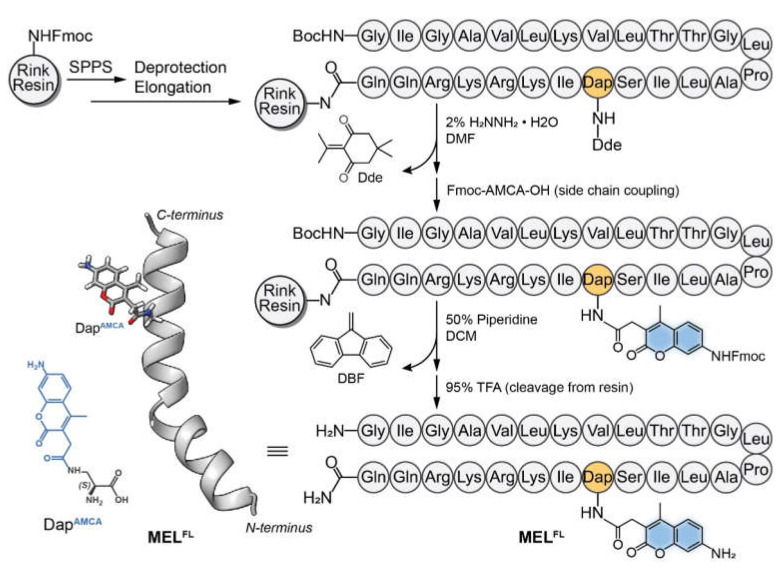
Structural illustration of MEL^FL^ monomer and solid-phase peptide synthesis of MEL^FL^ using Fmoc chemistry. The MEL^FL^ precursor MEL-Dde was synthesized de novo from Fmoc-Rink Resin by solid-phase peptide synthesis (SPPS). Dde was removed from MEL-Dde by adding an appropriate volume of 2% hydrazine hydrate/DMF solution. The product was mixed with Fmoc-AMCA-OH for side-chain coupling to obtain Fmoc-MEL^FL^-Rink Resin. Finally, Fmoc- and Rink Resin were removed to obtain MEL^FL^ by adding 50% piperidine/DCM and 95% TFA, respectively.

**Figure 2 toxins-14-00428-f002:**
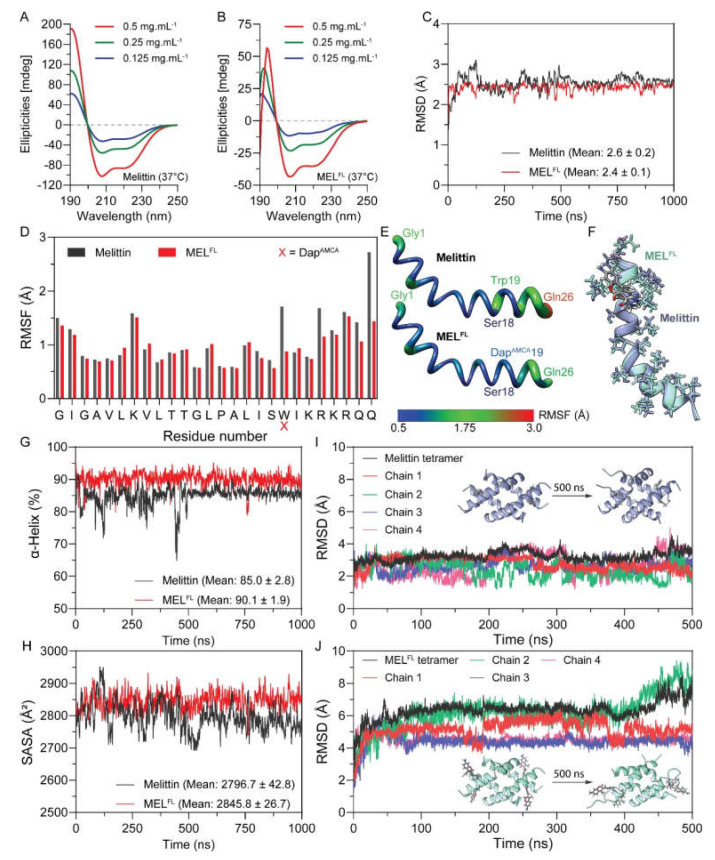
Structure characterizations and MD simulations of melittin and MEL^FL^. CD spectra of melittin (**A**) and MEL^FL^ (**B**) with different concentrations at 37 °C in 50% (*v/v*) TFE/water cosolvent mixtures. Data points represent the average of multiple independent experiments (*n* = 3). A comparative root mean square deviation (RMSD) graph of melittin and MEL^FL^ showing trajectories during 1 μs of MD simulation (**C**). The root mean square fluctuation (RMSF) analysis of all residues of melittin and MEL^FL^ (**D**). Backone representations mapping with the residue RMSF of the melittin and MEL^FL^ structure during the MD simulation period of 1 μs (**E**); the color bar indicates the range of RMSF values between 0.5 and 3.0. A structural alignment of the clustered central structure of melittin and MEL^FL^ was sampled from 1 μs of MD simulation (**F**). The dynamic α−helicity analysis of melittin and MEL^FL^ over the MD simulation period of 1 μs (**G**). Solvent accessible surface area (SASA) of melittin and MEL^FL^ as a function of time from the MD simulations (**H**). The 500 ns evolution of the RMSD profiles for melittin tetramer (**I**) and MEL^FL^ tetramer (**J**); the inserted snapshots show the initial and final packing states of protein tetramers.

**Figure 3 toxins-14-00428-f003:**
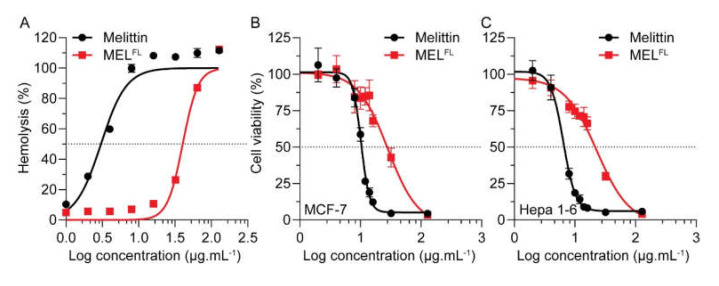
Hemolytic and therapeutic activity of melittin and MEL^FL^. Dose-dependent hemolytic activity of melittin and MEL^FL^. The HC_50_ of melittin and MEL^FL^ were determined to be 3.03 ± 0.02 and 40.30 ± 0.04 μg·mL^−1^ (*n* = 3), respectively (**A**). Cell-viability assays of MCF-7 (**B**) and Hepa 1-6 (**C**) cell lines treated with different concentrations of melittin/MEL^FL^ for 24 h. Corresponding IC_50_ values of melittin and MEL^FL^ are represented in Table 2.

**Figure 4 toxins-14-00428-f004:**
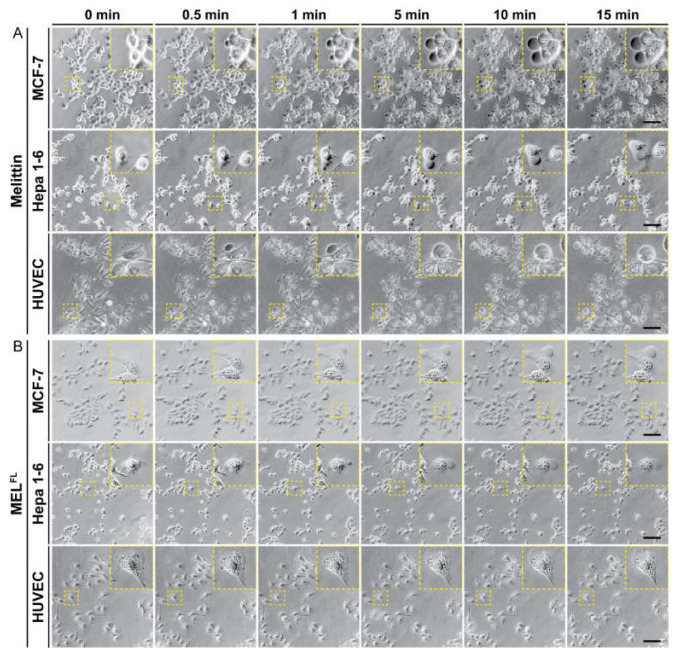
Cell morphological changes under optical microscopy in cells incubated with melittin and MEL^FL^. Melittin (10 μg·mL^−1^) (**A**) and MEL^FL^ (30 μg·mL^−1^) (**B**) treated MCF-7, Hepa 1–6, and HUVEC cell lines, with images captured at multiple time points. The yellow dashed line boxes and zoomed-in insets show morphological changes in the cell membrane. Data are representative of three independent experiments. Scale bar, 100 μm.

**Figure 5 toxins-14-00428-f005:**
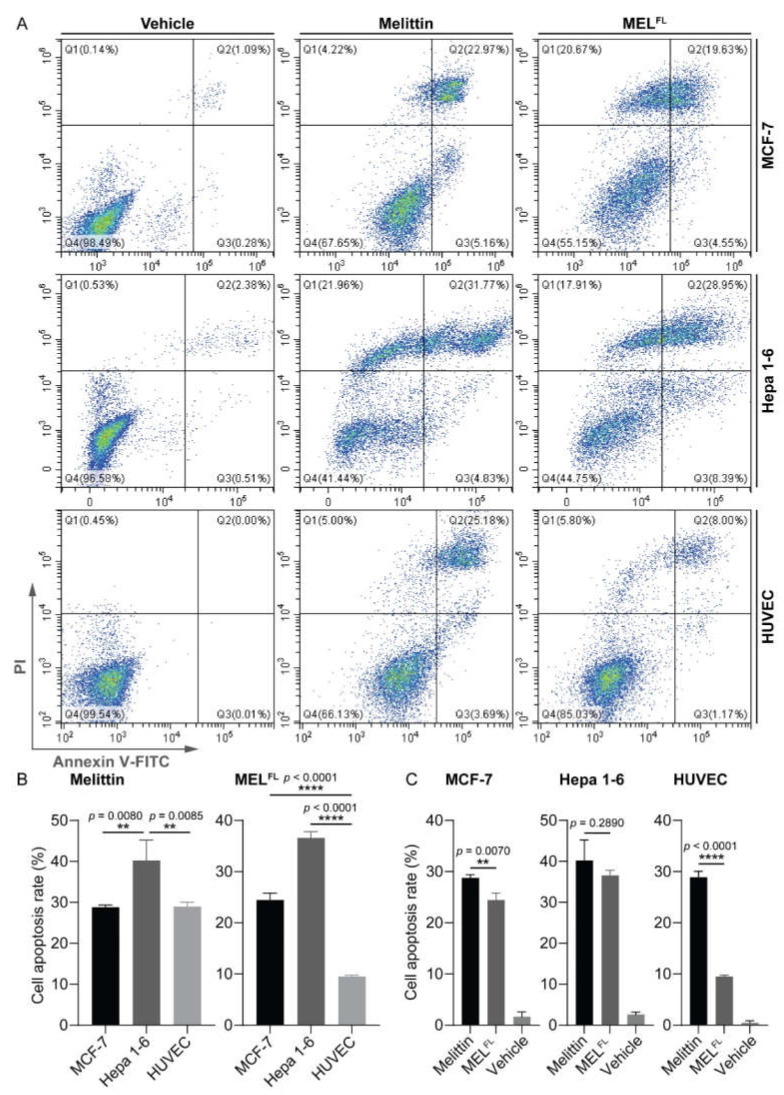
Flow cytometry apoptosis analysis of cancerous and normal cells treated with an IC_50_ (10 μg·mL^−1^) of melittin and 30 μg·mL^−1^ of MEL^FL^ for 2 h (*n* = 3 per group). The dot plots of Annexin V-FITC/PI FCM of MCF-7, Hepa 1-6, and HUVEC cells. The graphs stand for typical results of cell apoptosis (total apoptosis = Quadrant 2 + Quadrant 3) (**A**). Bar charts show the comparison of the percentage of total apoptosis between cells treated with melittin and MEL^FL^ in three different cell lines (**B**). The comparisons between two treatments (melittin and MEL^FL^) in the same cell lines (**C**). Results are presented as mean ± SEM. Differences among the three groups of cell lines were tested by one-way ANOVA. The comparisons between the two treatments (melittin and MEL^FL^) in the same cell line were carried out using Student’s *t*-test. ** *p* < 0.01; **** *p* < 0.0001.

**Figure 6 toxins-14-00428-f006:**
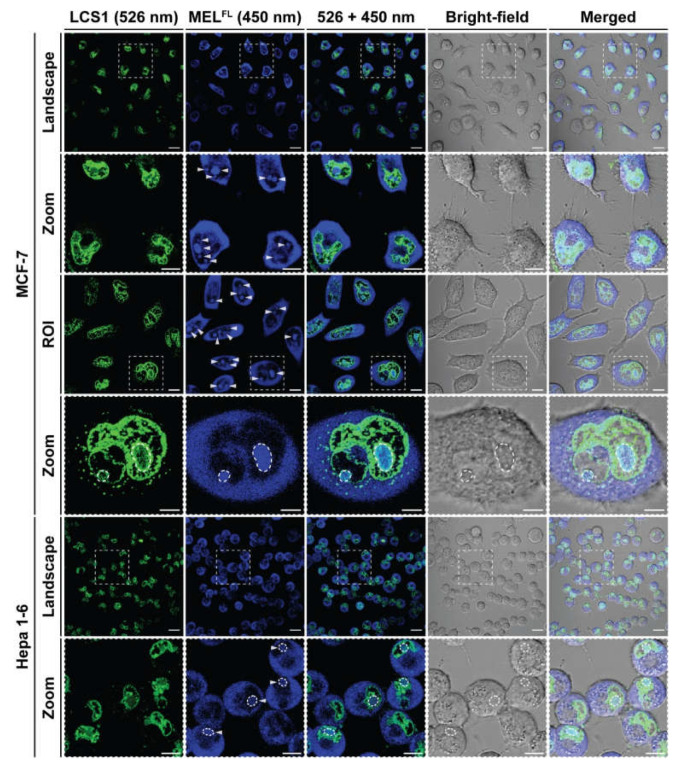
Confocal microscopy images of MCF-7 and Hepa 1-6 cells incubated with MEL^FL^ at 30 μg·mL^−1^. The cell nuclei were stained by Nuclear Green LCS1 for 20 min in the dark and subcellular localization of MEL^FL^ fluorescence was assessed after 2 h of incubation. ROI, region of interest. Scale bar, 10 μm.

**Table 1 toxins-14-00428-t001:** Relationship between peptide concentration and α-helix content.

Peptide	Concentration (mg·mL^−1^)	α-Helix (%) *	α-Helix (%) †	Mean (%)
	0.5	92.0	60.4	76.2
Melittin	0.25	99.3	65.9	82.6
	0.125	112.7	76.3	94.5
	0.5	47.4	30.0	38.7
MEL^FL^	0.25	49.3	31.5	40.4
	0.125	49.4	31.6	40.5

* α-helix% = (−[Ѳ]_208_ + 4000)/(33,000–4000) × 100; † α-helix% = (−[Ѳ]_222_ + 3000)/36,000 × 100.

**Table 2 toxins-14-00428-t002:** Therapeutic activity of peptides in cancerous cells.

Cell line	Peptide	IC_50_ (μg·mL^−1^)	SI
MCF-7	Melittin	10.22 ± 0.03	0.30
	MEL^FL^	28.01 ± 1.30	1.44
Hepa 1-6	Melittin	6.39 ± 0.15	0.47
	MEL^FL^	22.92 ± 0.37	1.76

SI denotes the selectivity index; SI = HC_50_/IC_50_.

## Data Availability

Not applicable.

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
