# Peer review of "Melittin Tryptophan Substitution with a Fluorescent Amino Acid Reveals the Structural Basis of Selective Antitumor Effect and Subcellular Localization in Tumor Cells"

_toxins, 2022, doi:10.3390/toxins14070428_

Round 1

Reviewer 1 Report

The authors work described in “Melittin tryptophan substitution with a fluorescent amino acid reveals the structural basis of selective antitumor effect and subcellular localization in tumor cells” show a unique look at the application of the honey bee derived melittin and exploration of a single Trp residue. The authors have a well-thought-out research design and approach to addressing the selectivity of the Trp residue. Moreover, the manuscript is well written and will be of use to the broader research communities in peptide design/probes-synthesis, venom based therapeutics, and pharmacology.

I don’t believe any major revisions are needed, but I have the following minor revisions.

  1. Supplemental Figure. Please fix this/organize it better than its current form. It looks messy, there is not enough detail (flowrate, amount injected, detection wavelength) on the HPLC and MS used for this analysis (please add it to the methods). A citation would suffice from the groups previous work.

  1. Figure 1. Figure Scheme beautifully illustrates the synthesis of melittin. Please provide a more detail in the figure caption.

  1. Figure 4. What is ROI? It is not defined and why was this not performed on the Hepa 1-6 cell line? Moreover, what is the overall percent population of cells that have MELfl? This would be nice to know the quantification of this as it would suggest the penetrance into the cells.

Minor revisions

 -Italics --        The in vitro hemolytic and anticancer activity assays reveal that introducing Line 10

-Do you mean “and”?  reduced hemolytic toxicity but improved the therapeutic index (TI), with up to a five-fold increase line 12

-Italics  in situ-- melittin and enables in situ fluorescent cellular tracking of melittin. The melittin analog exhibited a 20

-Italics and consistency in situ--In the current study, we report the in-situ visualizations and investigation of fluores-52

Italics  in vitro --Following structural and in vitro safety analysis, the human and mouse cancer-line 63

Italics in vitro --cell lines MCF-7 and Hepa 1-6 were investigated as in vitro model systems to observe line 64

Space after 500- peptides' conformation reached a steady state in the last 500ns, and the MELFL exhibited line 137

Necrotic cells, please justify (PI is not spelled out and no mention it is used for necrosis---dot plot (Figure 3D-I) was divided into four quadrants as necrotic cells (Q1, FITC-/PI+), line 202

Necrosis, please justify (Figure 3G-I). The elevated necrosis was also observed after treatment of melittin and line  209

-Do you mean “and”?  duced hemolytic activity but greatly improved the therapeutic index (TI) compared to line  249

What temp? phere of 5% CO2 and 90% relative humidity. In order to maintain a good growth condition line 281

Cannot start sentences with a numerical value, Spell it out -250nm was recorded (step 1nm) at 37°C. 50% TFE buffer was measured as a control. Line 292

Cannot start sentences with a numerical value , Spell it out --Dong et al [39]. 1 mL of venous blood of New Zealand rabbits from the ear veins was line 300

Space after 2---pended in 2mL sterile PBS, and then centrifuged again under the same conditions to dis-line 305

1 or 2 ? in the logarithmic phase were seeded into 96-well plates at 1-2 x 104 per well and then line 320

Cannot start sentences with a numerical value, Spell it out-- moved. 150 μL of dimethyl sulfoxide (DMSO) was added to each well and shaken at room line 324

 Hepa 1-6 ? and Heap 1-6 were seeded into a 6-well plate and cultured in DMEM complete medium line 334

 Use constant units --at a density of 1.0*105 cells/well at 37°C and 5% CO2 overnight to fully adhere to the wall. Line 335

Cannot start sentences with a numerical value, Spell it out--solvated using equal volume ratios of water and trifluoroethanol. 6 chloride ions were line 363

You mean Significant? --be statistically different. Line 382

Author Response

Response to Reviewer 1 Comments

The authors work described in “Melittin tryptophan substitution with a fluorescent amino acid reveals the structural basis of selective antitumor effect and subcellular localization in tumor cells” show a unique look at the application of the honey bee derived melittin and exploration of a single Trp residue. The authors have a well-thought-out research design and approach to addressing the selectivity of the Trp residue. Moreover, the manuscript is well written and will be of use to the broader research communities in peptide design/probes-synthesis, venom based therapeutics, and pharmacology.

I don’t believe any major revisions are needed, but I have the following minor revisions.

Author response: We thank the reviewer for his positive comment.

Point 1: Supplemental Figure. Please fix this/organize it better than its current form. It looks messy, there is not enough detail (flowrate, amount injected, detection wavelength) on the HPLC and MS used for this analysis (please add it to the methods). A citation would suffice from the groups previous work.

Response 1: Thank you for pointing this out. The Supplemental Figures have been corrected by adding details for HPLC and MS analyses, where the change can be found in the uploaded attachment.

Point 2: Figure 1. Figure Scheme beautifully illustrates the synthesis of melittin. Please provide a more detail in the figure caption.

Response 2: The suggested correction has been made. The figure caption now reads:

“Figure 1. Scheme of solid-phase peptide synthesis of MELFL using Fmoc chemistry. The MELFL precursor MEL-Dde was synthesized de novo from Fmoc-Rink Resin by solid-phase peptide synthesis (SPPS). Dde was removed from MEL-Dde by adding an appropriate volume of 2% hydrazine hydrate/DMF solution. The product was mixed with Fmoc-AMCA-OH for side-chain coupling to obtain Fmoc-MELFL-Rink Resin. Finally, Fmoc- and Rink Resin were removed to obtain MELFL by adding 50% piperidine/DCM and 95% TFA, respectively.” The change can be found in the uploaded revised manuscript.

Point 3: Figure 4. What is ROI? It is not defined and why was this not performed on the Hepa 1-6 cell line? Moreover, what is the overall percent population of cells that have MELfl? This would be nice to know the quantification of this as it would suggest the penetrance into the cells.

Response 3: Thank you for pointing this out. The ”ROI” stands for a region of interest, and the annotation was added in the Figure 4 caption. The missing ROI indication (dash line circles) was added in the corrected figure. With regards to the overall percentage of cells with MELFL uptake, the observation using confocal microscopy found that MELFL could easily penetrate each cell and abundantly distributed in the cell membrane and nucleus. The uneven color of the cell slices in the microscope picture is mainly due to the unequal level of cells in different regions.

Point 4: -Italics --        The in vitro hemolytic and anticancer activity assays reveal that introducing Line 10

Response 4: We have changed the “in vitro” into “in vitro”.

Point 5: -Do you mean “and”?  reduced hemolytic toxicity but improved the therapeutic index (TI), with up to a five-fold increase line 12

Response 5: Fixed.

Point 6: -Italics  in situ-- melittin and enables in situ fluorescent cellular tracking of melittin. The melittin analog exhibited a 20

Response 6: We have changed the “in situ” into “in situ”.

Point 7: -Italics and consistency in situ--In the current study, we report the in-situ visualizations and investigation of fluores-52

Response 7: We have changed the “in situ” into “in situ”.

Point 8: Italics  in vitro --Following structural and in vitro safety analysis, the human and mouse cancer-line 63

Response 8: We have changed the “in vitro” into “in vitro”.

Point 9: Italics in vitro --cell lines MCF-7 and Hepa 1-6 were investigated as in vitro model systems to observe line 64

Response 9: We have changed the “in vitro” into “in vitro”.

Point 10: Space after 500- peptides' conformation reached a steady state in the last 500ns, and the MELFL exhibited line 137

Response 10: We have revised it accordingly.

Point 11: Necrotic cells, please justify (PI is not spelled out and no mention it is used for necrosis---dot plot (Figure 3D-I) was divided into four quadrants as necrotic cells (Q1, FITC-/PI+), line 202

Response 11: We have provided an annotation for “PI” in the Figure 3 caption. Q1 represents the nuclear fragments or necrotic cells stained with only PI. The correction has been made to prevent the misleading.

Point 12: Necrosis, please justify (Figure 3G-I). The elevated necrosis was also observed after treatment of melittin and line  209

Response 12: See the response to point 11.

Point 13: -Do you mean “and”?  duced hemolytic activity but greatly improved the therapeutic index (TI) compared to line  249

Response 13: Thank you for pointing this out. We have changed it accordingly.

Point 14: What temp? phere of 5% CO2 and 90% relative humidity. In order to maintain a good growth condition line 281

Response 14: Thank you for pointing this out. We have added the temperature (37℃) in section 5.3.

Point 15: Cannot start sentences with a numerical value, Spell it out -250nm was recorded (step 1nm) at 37°C. 50% TFE buffer was measured as a control. Line 292

Response 15: Thank you for pointing this out. The sentence now reads: “Fifty percent of TFE buffer was measured as a control.”

Point 16: Cannot start sentences with a numerical value , Spell it out --Dong et al [39]. 1 mL of venous blood of New Zealand rabbits from the ear veins was line 300

Response 16: Thank you for pointing this out. The sentence now reads: “One milliliter of venous blood of New Zealand rabbits…”

Point 17: Space after 2---pended in 2mL sterile PBS, and then centrifuged again under the same conditions to dis-line 305

Response 17: Fixed.

Point 18: 1 or 2 ? in the logarithmic phase were seeded into 96-well plates at 1-2 x 104 per well and then line 320

Response 18: Thank you for pointing this out. We have revised it accordingly.

Point 19: Cannot start sentences with a numerical value, Spell it out-- moved. 150 μL of dimethyl sulfoxide (DMSO) was added to each well and shaken at room line 324

Response 19: Thank you for pointing this out. The sentence now reads: “Then, 150 μL of dimethyl sulfoxide (DMSO) was added…”

Point 20: Hepa 1-6 ? and Heap 1-6 were seeded into a 6-well plate and cultured in DMEM complete medium line 334

Response 20: Thank you for pointing this out. We have revised it accordingly. See Section 5.7.

Point 21: Use constant units --at a density of 1.0*105 cells/well at 37°C and 5% CO2 overnight to fully adhere to the wall. Line 335

Response 21: Thank you for pointing this out. We have revised it accordingly. See Section 5.7.

Point 22: Cannot start sentences with a numerical value, Spell it out--solvated using equal volume ratios of water and trifluoroethanol. 6 chloride ions were line 363

Response 22: Thank you for pointing this out. The sentence now reads: “Six chloride ions were…”

Point 23: You mean Significant? --be statistically different. Line 382

Response 23: Thank you for pointing this out. The sentence now reads: “P-value less than 0.05 was considered to be statistically significant.”

Reviewer 2 Report

The article titled: "Melittin tryptophan substitution with a fluorescent amino acid reveals the structural basis of selective antitumor effect and
subcellular localization in tumor cells"; investigated the effect of the substitution of Trp19 with a noncanonical fluorescent amino acid (DapAMCA) in cell viability of tumoral cell lineages and hemolytic activity, the authors suggest that this modification reduce the unspecific toxic effect (hemolysis) but preserve the antitumoral properties (inducing cell death of tumoral cell lineages). They also suggest that the modification alters the capacity of the molecule to migrate to nuclear and nucleolar targets.   However, the results presented does not support the antitumoral capacity of melittin neither the mechanism by which the authors proposed.

The major consideration is the fact that the authors assume that the aminoacid substitution improve the therapeutic index based in the fact that the hemolytic effect reduces in comparison with native melittin. However, the aminoacid substitution increases significatively the IC50 (table 2). The use of therapeutic index in this case seems not be correctly employed, since the lytic effect are being reported in rabbit erythrocytes not in the cancer cell lineages. In this sense, there are missing experiments evaluating the lytic effect in the cancer cell lineages. It is important also to evaluate an in vivo effect should using experimental models to confirm the anticancer effect of modified melittin and to confirm the advantages of MELfl in comparison with native mellitin. Besides that, in the flow cytometry analyses the author did not provide the statistical comparison of viable (PIneg/AnnexinVneg); Early apoptotic (PIneg/Annexin V +) and necrotic/late apoptotic populations (double positive) from Mel and MelFl. Furthermore, the confocal microscopy analysis just showed that MELFL is localized in the nuclei, however there are no comparison with native form of melittin to demonstrate that the native form has a different localization. Finally, there is no experiments that proves that nuclear localization of MELfl is related with their supposed anticancer effect.

In this sense, the present work must be improved to be able to be published, specially with further analysis to confirm or refute the hypothesis of the anticancer properties of MELfl and their advantages in comparison with native melittin.

Author Response

Response to Reviewer 2 Comments

The article titled: "Melittin tryptophan substitution with a fluorescent amino acid reveals the structural basis of selective antitumor effect and

subcellular localization in tumor cells"; investigated the effect of the substitution of Trp19 with a noncanonical fluorescent amino acid (DapAMCA) in cell viability of tumoral cell lineages and hemolytic activity, the authors suggest that this modification reduce the unspecific toxic effect (hemolysis) but preserve the antitumoral properties (inducing cell death of tumoral cell lineages). They also suggest that the modification alters the capacity of the molecule to migrate to nuclear and nucleolar targets.   However, the results presented does not support the antitumoral capacity of melittin neither the mechanism by which the authors proposed.

Author response:

We would like to thank the reviewer for the careful and thorough reading of this manuscript and the thoughtful comments and constructive suggestions, which helped improve the quality of this manuscript. Our response follows

Point 1: The major consideration is the fact that the authors assume that the aminoacid substitution improve the therapeutic index based in the fact that the hemolytic effect reduces in comparison with native melittin. However, the aminoacid substitution increases significatively the IC50 (table 2).

Response 1: Thank you for pointing this out. Although the IC50 of MELFL was increased by 3-fold, the fact is that TI of MELFL was increased by 4.8-fold in MCF-7 as compared to the melittin. This suggests MELFL confers increased tolerance to therapy.

Point 2: The use of therapeutic index in this case seems not be correctly employed, since the lytic effect are being reported in rabbit erythrocytes not in the cancer cell lineages. In this sense, there are missing experiments evaluating the lytic effect in the cancer cell lineages.

Response 2: we agree that more in vitro evidence is required to support the selectivity of the modified melittin, and we are asking the editor to extend the round 2 deadline for conducting the in vitro selectivity assay in both non-cancerous control cell line and cancer cell lines. In addition, we consider the term “therapeutic index” used in the manuscript may not be appropriate, the “selectivity index (HC50/IC50)” may be more suitable to reflect the utility, and has been reported in numerous reports (e.g., DOI: 10.3390/molecules22020207; DOI: 10.1038/s42003-020-01261-0).

Point 3: It is important also to evaluate an in vivo effect should using experimental models to confirm the anticancer effect of modified melittin and to confirm the advantages of MELfl in comparison with native mellitin.

Response 3: While we appreciate the reviewer’s feedback, we respectfully disagree. We think the primary goal of this study is to develop a labeling method to study the mechanistic role of the crucial residue (tryptophan) without altering the helical structure of melittin. Such fluorescent labeling allows us to monitor the modified melittin (MELFL) close to its native way at the cellular level. However, the in vivo demonstration of the advantages of MELFL is beyond the scope of this manuscript because the purpose of Trp−19−DapAMCA substitution was not primarily designed to improve the therapeutic capacity in anticancer treatment.

Point 4: Besides that, in the flow cytometry analyses the author did not provide the statistical comparison of viable (PIneg/AnnexinVneg); Early apoptotic (PIneg/Annexin V +) and necrotic/late apoptotic populations (double positive) from Mel and MelFl.

Response 4: Thank you for pointing this out. We are now performing additional flow cytometry analyses to provide the statistical comparison of viables. We will provide these data in the round 2 revision.

Point 5: Furthermore, the confocal microscopy analysis just showed that MELFL is localized in the nuclei, however there are no comparison with native form of melittin to demonstrate that the native form has a different localization.

Response 5: We unfortunately cannot provide a demonstration to localize the native melittin, this the reason why we developed this labeling strategy to investigate the role of tryptophan.

Point 6: Finally, there is no experiments that proves that nuclear localization of MELfl is related with their supposed anticancer effect.

Response 6: Thank you for pointing this out. It would have been interesting to explore this aspect. However, in the case of our study, it seems slightly out of scope because the anticancer effects of melittin are multifaceted (DOI: 10.1007/s00280-016-3160-1). We think this study makes a valuable contribution that melittin has the nuclear targeting ability of MELFL implies that the melittin may be modified for nuclear and nucleolar targeted therapy.

Point 7: In this sense, the present work must be improved to be able to be published, specially with further analysis to confirm or refute the hypothesis of the anticancer properties of MELfl and their advantages in comparison with native melittin.

Response 7: We agree with the reviewer’s suggestions. The relative analyses are being conducted. and we are asking the editor to extend the round 2 deadline so that we can revise the manuscript more appropriately.

Reviewer 3 Report

            The article “Melittin tryptophan substitution with a fluorescent amino acid reveals the structural basis of selective antitumor effect and subcellular localization in tumor cells” is a very interesting study. Briefly, it describes the effects of the substitution of Trp19 with a  fluorescent amino acid in the melittin molecule. 

             The article was pleasantly conducted, with elegant assays. Still, the results were well exposed/explained and the English is almost impeccable. It has been a while since I've reviewed an article like this.

               During my reading, I was wondering if the W replacement would affect the tetramer, which was answered, and I agree that it would be the main justification for the reduction of hemolysis. 

               My biggest question and criticism of the article is that the authors should have conducted the biological cancer assays also with a non-cancerous control cell line, to compare the results and really suggest if the modified molecule could be used for a future anticancer therapy. Melittin can kill cancer cell lines, bacterias, virus, and also health cells, that is why it is not used in therapy so far. Maybe, if it is not possible to do this now; however, at least something should be added in the discussion.

Minor issues:

Abstract: Please be consistent: in vitro in italics or not. Do the same for in situ in key contribution. See also line 52 (in-situ is with a dash). Also lines: 59, 63, etc. 

Line 31: add comma after antiviral (Oxford comma)

I also recommend the read and maybe cite the articles below:

  • 10.3389/fphar.2020.00611
  • 10.3389/fimmu.2019.02090  

Author Response

Response to Reviewer 3 Comments

The article “Melittin tryptophan substitution with a fluorescent amino acid reveals the structural basis of selective antitumor effect and subcellular localization in tumor cells” is a very interesting study. Briefly, it describes the effects of the substitution of Trp19 with a  fluorescent amino acid in the melittin molecule.

The article was pleasantly conducted, with elegant assays. Still, the results were well exposed/explained and the English is almost impeccable. It has been a while since I've reviewed an article like this.

During my reading, I was wondering if the W replacement would affect the tetramer, which was answered, and I agree that it would be the main justification for the reduction of hemolysis.

My biggest question and criticism of the article is that the authors should have conducted the biological cancer assays also with a non-cancerous control cell line, to compare the results and really suggest if the modified molecule could be used for a future anticancer therapy. Melittin can kill cancer cell lines, bacterias, virus, and also health cells, that is why it is not used in therapy so far. Maybe, if it is not possible to do this now; however, at least something should be added in the discussion.

Author response: We thank the reviewer for his positive comment. We agree with the reviewer’s concerns, and we are asking the editor to extend the round 2 deadline for conducting the in vitro selectivity assay in a non-cancerous control cell line.

Minor issues:

Point 1: Abstract: Please be consistent: in vitro in italics or not. Do the same for in situ in key contribution. See also line 52 (in-situ is with a dash). Also lines: 59, 63, etc.

Response 1: Thank you for pointing this out. Reviewer 1 also mentioned this issue. We have corrected them accordingly, where the change can be found in the revised manuscript.

Point 2: Line 31: add comma after antiviral (Oxford comma)

Response 2: Fixed.

Point 3: I also recommend the read and maybe cite the articles below:

10.3389/fphar.2020.00611

10.3389/fimmu.2019.02090

Response 2: Thank you for recommending these two important articles. We have carefully read and appropriately inserted them into the manuscript (references 6 and 7).

Round 2

Reviewer 2 Report

Unfortunantly, the authors did not presented the experiments that should be done to prove the main hypothesis of this investigation. In this sense, I considered that the paper must be improoved to achieve the proper quality to be published.

Round 3

Reviewer 2 Report

The authors significantly improved the article as recommended. However, the following points should be modified and explained as listed below.

Major revisions 

Since the hemolysis assays used venous blood of New Zealand rabbits the number of Ethical Committee approval must be provided in the Institutional Review Board Statement. Describe in the Material and methods section the sanitary and environmental conditions in which these animals were kept.

In the MTT assay was used the wavelength of 490nm (line 475) is not usually used, since the maximum absorbance of formazan is in 570 nm. Please provide an explanation with references to justify the use this wavelength.

Line 260 - the quadrant 1 was defined as necrotic cells, however necrotic cells present also positivity Annexin-V, therefore this quadrant should not be included in the analysis, please check the data.

In the figure 3 the resolution and magnification are not adequate to observe the changes described by the authors, please improve the images. The information of how many experiments or microscope fields the images are representing, should be provided.

In the conclusion line 374 the authors conclude that the Trp19 residue substitution in melittin enable selective anticancer activity. However, there is no sufficient experimental evidence to confirm this hypothesis, we only can conclude, by the data presented that this modification reduces the hemolytic activity of melittin.      

Minor revisions 

Figure 1 should be placed at material and methods section

Figure 4 - In the bar graphs insert the cell apoptosis rate of the controls (vehicle treated) and define in the legend text how this rate was calculated (Quadrant 2 + Quadrant 3). 

Line 70 - change "in vitro model systems to observe anticancer effect..." were used to evaluate the effect in the viability of cancer cell lines... Since "anticancer effect is not a correct term to use in an invitro model" 

Line 174 the same modification of line 70 should be done 

Round 4

Reviewer 2 Report

Once the authors made all the changes that were requested and responded adequately to the questions, I believe that the article is suitable for publication.